# Semi-Supervised Active Learning for Object Detection

**Sijin Chen, Yingyun Yang * and Yan Hua** 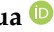

School of Information and Communication Engineering, Communication University of China,
Beijing 100024, China
* Correspondence: yangyingyun@cuc.edu.cn

**Abstract:** Behind the rapid development of deep learning methods, massive data annotations are indispensable yet quite expensive. Many active learning (AL) and semi-supervised learning (SSL) methods have been proposed to address this problem in image classification tasks. However, these methods face a new challenge in object detection tasks, since object detection requires classification as well as localization information in the labeling process. Therefore, in this paper, an object detection framework combining active learning and semi-supervised learning is presented. Tailored for object detection tasks, the uncertainty of an unlabeled image is measured from two perspectives, namely classification stability and localization stability. The unlabeled images with low uncertainty are manually annotated as the AL part, and those with high uncertainty are pseudo-labeled with the detector's prediction results as the SSL part. Furthermore, to better filter out the noisy pseudo-boxes brought by SSL, a novel pseudo-label mining strategy is proposed that includes a stability aggregation score (SAS) and dynamic adaptive threshold (DAT). The SAS aggregates the classification and localization stability scores to measure the quality of predicted boxes, while the DAT adaptively adjusts the thresholds for each category to alleviate the class imbalance problem. Extensive experimental results demonstrate that our proposed method significantly outperforms state-of-the-art AL and SSL methods.

**Keywords:** active learning; semi-supervised learning; object detection; stability-based sample selection; pseudo-label mining strategy





## 1. Introduction

With the rapid development of deep learning, many computer vision tasks have achieved significant improvements, such as image classification, object detection, etc. Behind these advances, plenty of annotated data play an important role. However, labeling accurate annotations for large-scale data is usually time-consuming and expensive, especially for object detection, which requires annotating precise bounding boxes for each instance, besides category labels. As the complexity of the scene and the number of target categories increase, the annotation cost continues to rise. Under such circumstances, there is an increasing demand for minimized labeling costs and the full use of large-scale unlabeled data in an incremental learning manner. Active learning (AL) [1–6] and semi-supervised learning (SSL) [7–9] are two popular methods to resolve this problem.

Active learning aims to selectively annotate the most informative unlabeled data to achieve high detection performance with minimum labeling costs. Learning-based AL methods select a batch of samples to label with guidance from the previously trained model and then add these samples into the labeled dataset for the model training in the next cycle. Although active learning has effectively reduced the labeling cost, it fails to take full advantage of the unlabeled data. Most accessible unlabeled samples are not utilized in active learning. Meanwhile, semi-supervised learning, which can learn feature representations with limited supervision by exploring the massive amount of unlabeled images, has trouble coping with extremely hard samples and suffers from the confirmation bias problem. In this paper, we combine active learning and semi-supervised learning for

object detection into one framework, so that active learning and semi-supervised learning can draw on each other's strengths to make full use of the semantic information of the unlabeled data.

In the existing literature, active learning methods [10–12] can roughly be divided into two categories, namely uncertainty-based methods and distribution-based methods. Uncertainty-based methods define various metrics to pick out the most uncertain samples based on the current model. Distribution-based methods [2,3] aim to pick out diverse samples to represent the distribution of a dataset. Although active learning methods are very popular for image classification, there are relatively few active learning methods specified for object detection tasks [13,14]. Classification-based methods only consider the predicted class distribution, while the bounding box prediction could be equally important for selecting informative samples in object detection. Therefore, directly applying active learning methods for image classification to object detection may lead to sub-optimal results. Tailored for object detection, a unified metric containing localization stability and classification stability to measure the uncertainty of an unlabeled image is proposed. We apply data augmentations to each unlabeled image and feed two different views of the same image to the detector. Then, the uncertainty of the image is calculated according to the detector's prediction results.

Semi-supervised learning methods employ easily accessible unlabeled data to facilitate model training with limited annotated data. Currently, there are two main approaches to achieve this goal: pseudo-labeling [15,16] and consistency regularization [17,18]. Pseudo-labeling is a technique that utilizes trained models to generate labels for unlabeled data. Meanwhile, the consistency-based regularization, from another perspective, forces a model to have similar output when giving a normal and a perturbed input with different data augmentations and perturbations such as dropout. In Semi-Supervised Object Detection (SSOD), some studies borrow the key techniques (e.g., pseudo-labeling, consistency training) and directly apply them to SSOD. The predicted boxes from a pre-trained "teacher" detector are used as the annotations of unlabeled images to train the "student" detector, where the same images are applied with different augmentations for the teacher and student model. It is common to apply confidence-based filtering to each predicted box (after NMS) with a high threshold value to filter out noisy boxes. While this method is effective, we argue that the pseudo-label mining strategy is sub-optimal for SSOD. The reasons are twofold. (1) the classification scores cannot reflect the localization quality of pseudo-boxes. In image classification, prediction scores naturally represent the likelihood of an object appearing in an image, and thus a confidence score is a reasonable choice. However, as detection involves localizing and classifying objects using two separate branches through regression and classification, filtering out boxes based on class predictions is not appropriate. (2) A static threshold could amplify the class imbalance problem. A model's accuracy for detecting different classes of objects usually varies. As a result, lower-confidence predictions from underrepresented classes are oftentimes filtered out with a fixed high threshold, which works well for top-performing classes.

To mitigate these issues, we propose a novel pseudo-label mining strategy tailored for object detection, including a stability aggregation score (SAS) and dynamic adaptive threshold (DAT). Specifically, the SAS replaces the confidence score with the stability aggregation score, which combines classification stability and localization stability to measure the overall quality of pseudo-boxes. Meanwhile, the DAT adaptively adjusts the thresholds on a per-category basis to alleviate the class imbalance problem.

In summary, a unit detection framework that combines active learning and semi-supervised learning is proposed to maximize the utilization of unlabeled data. Customized for object detection tasks, the uncertainty of unlabeled images is measured from two aspects: classification stability and localization stability. Furthermore, we design a novel pseudo-label mining strategy that includes a stability aggregation score (SAS) and dynamic adaptive threshold (DAT) to better filter out the noisy pseudo-boxes. Extensive experiments show that our method outperforms state-of-the-art AL and SSL methods.

## 2. Related Work

**Active Learning.** There exist plenty of active learning methods for image classification [19–21]. The most popular methods are based on pool-based selective sampling. Most active learning methods can also be classified under two categories: uncertainty-based methods and distribution-based methods. The uncertainty-based methods use the posterior probability of the predicted class [22,23] or the margin between the posterior probabilities of the first and the second predicted class [24,25] to figure out the informative unlabeled samples. It can also be defined upon entropy [26,27] to measure the variance of unlabeled samples. Distribution-based methods select diverse samples by estimating the layout of unlabeled samples. Core-set [2] defines active learning as core-set selection, i.e., choosing a set of points such that the model trained on the labeled set can capture the diversity of the unlabeled set. There are also methods that consider uncertainty and distribution in conjunction. In addition to image classification tasks, active learning has also been explored in other vision tasks, such as segmentation [28] and 3D object detection [29]. Compared to image classification tasks, there are relatively few active learning methods specified for object detection tasks. By simply sorting the loss predictions of instances to evaluate the image uncertainty, the learning loss method designed for image classification was directly applied to object detection. SSM [13] uses a copy–paste strategy to cross-validate the uncertainty of images. The work [14] introduces two methods: localization tightness with the classification information (LT/C) and localization stability with the classification information (LS+C). The former is based on the overlapping ratio between the region proposals and the final prediction, which can only be applied to two-stage detectors. The latter is based on the variation in predicted object locations when input images are corrupted by noise, which neglects the classification branch.

**Semi-Supervised Learning.** SSL methods leverage unlabeled images for improved performance in various tasks [30–32]. Semi-supervised learning can be categorized into two groups: pseudo-labeling and consistency training. Pseudo-labeling methods improve the model's performance by generating high-quality pseudo-labels of unlabeled data and retraining the model. Noisy Student Training [33] injects noise into unlabeled data training, which equips the model with stronger generalization through training on a combination of labeled and unlabeled data. Mean Teacher [17] uses the Exponential Moving Average (EMA) of the student as the teacher model to conduct online pseudo-labeling. Based on the assumption that the model should be invariant to small changes in input images, consistency training forces the model to make similar predictions on the perturbed versions of the same image. UDA [18] validates the crucial role that advanced data augmentations play in consistency training. Some works also consider combining pseudo-labeling and consistency training to achieve better performance. FixMatch [8] applies weak and strong augmentations to the same image and uses the weakly augmented version to generate pseudo-labels. The model is trained on strongly augmented versions to align predictions with pseudo-labels.

Semi-supervised learning has transferred a great deal of experience to the SS-OD domain [34–38]. STAC [37] generates pseudo-labels on unlabeled data using a static teacher model trained on labeled images. These pseudo-labels are then selected and used for training along with the labeled data. After STAC, Instant Teaching borrows the idea of Exponential Moving Average (EMA) from Mean Teacher, and updates the teacher model after each training iteration to generate instant pseudo-labels, realizing an end-to-end framework. Instant Teaching [39] also introduces a model ensemble to aggregate predictions from multiple teacher models to overcome confirmation bias problems. Though promoting the quality of pseudo-labels, ensemble methods introduce considerable computation overhead. Unbiased Teacher [40] replaces the traditional cross-entropy loss with focal loss to alleviate the class-imbalanced pseudo-labeling issue, showing strong performance when labeled data are scarce.

## 3. Materials and Methods

### 3.1. Overall Pipeline

Aiming at improving the performance of the detection model where a set of labeled images with instance-level annotations and a set of unlabeled images are used for training, two popular learning methods (AL and SSL) are aggregated into a single framework. The overall pipeline of the proposed framework is depicted in Figure 1. In step 1, the labeled part of all data is utilized to train a detection model $M_t$ in a supervised manner. In step 2, the stability of the unlabeled data is calculated by the detection model $M_t$ to obtain a stability set of the unlabeled dataset U. As the AL part, low-stability samples are manually annotated, which cannot be properly represented by SSL. Meanwhile, as the SSL part, we perform a pseudo-annotation mining strategy to discover stable predictions in the prediction result set and pseudo-label them. Manual annotations and pseudo-annotations, along with the corresponding unlabeled images, are then added into the labeled dataset D to facilitate the detection model's performance.

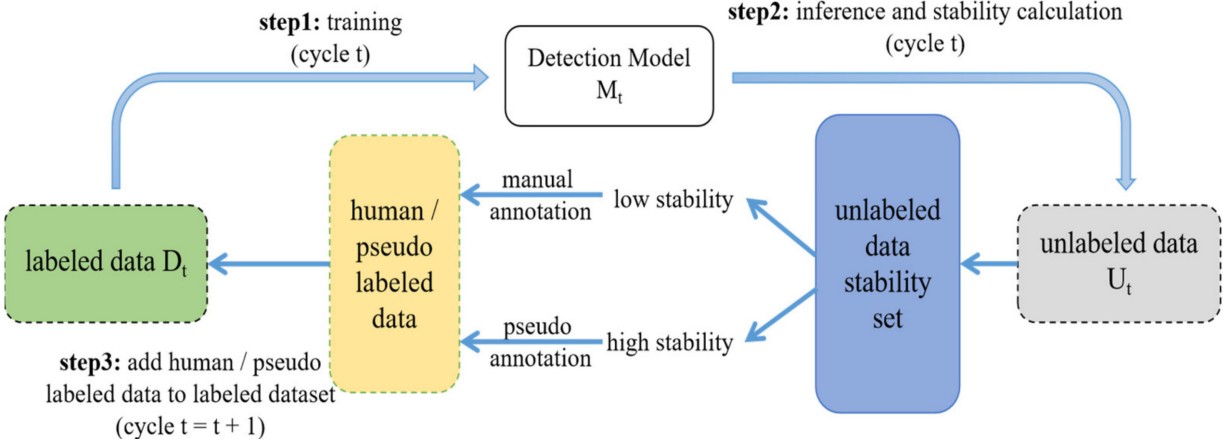

**Figure 1.** Illustration of the proposed framework at the t-th cycle.

For a better understanding and analysis, we formulate the proposed semi-supervised active learning process as follows.

Let $D = \{(x_i, y_i)\}_{i=1}^{N}$ denote the labeled dataset D, where N is the number of images in labeled dataset D and $x_i$, $y_i$ represent an image and corresponding label information, respectively. For the object detection task, the annotation information $y_i$ can be represented as $y_i = \{Y_k, B_k\}_{k=1}^{K}$, indicating that image $x_i$ contains K instances; the k-th annotation includes one-hot type classification label $Y_k$ and bounding box label $B_k$ containing coordinates of the top left corner (x1, y1) and coordinates of the lower right corner (x2, y2).

Let $U = \{x_j\}_{j=1}^{N_u}$ denote unlabeled images, where $N_u$ is the number of unlabeled images in unlabeled dataset U. In a semi-supervised active learning cycle t, the algorithm selects $X_t^{al}$ and $X_t^{ssl}$ from the unlabeled dataset $U$, respectively, where $X_t^{al}$ represents a manually labeled dataset for active learning, and $X_t^{ssl}$ represents a pseudo-labeled dataset for semi-supervised learning. Meanwhile, the labeled dataset is updated from $D_t$ to $D_{t+1}$, where $D_{t+1}$ denotes the sum of datasets $D_t$, $X_t^{al}$, and $X_t^{ssl}$. Similarly, the unlabeled dataset $U_t$ is updated to $U_{t+1}$, where $U_{t+1}$ denotes $U_t$ without selected datasets $X_t^{al}$ and $X_t^{ssl}$.

The framework depicted above consists of two key components: the active learning stability calculation methodology and the semi-supervised learning pseudo-annotation mining strategy. In the following sections, these two main components will be explained in detail.

### 3.2. AL Sampling Strategy

From the pipeline of the proposed framework in Figure 1, we can see that the most important part of the active learning process is the definition of "informative samples". Tailored for the object detection task, we design an AL sampling strategy based on the predictive stability of instances. The strategy includes four steps: (1) model inference prediction, (2) predicted box matching, (3) localization and classification stability calculation, and (4) determination of sampling metrics. The details of each step are described below.

#### 3.2.1. Model Inference Prediction

For an unlabeled image $x_u$ in an unlabeled dataset, we denote the augmented view of $x_u$ as $x_u^{'} = A(x_u)$, where $A(\cdot)$ denotes the augmentation conducted on image $x_u$. $x_u$ and $x_u^{'}$ enter into detection model M with parameters $\theta$ to obtain the corresponding results:

$$\{(b_i, p_i)\} = M(x_u; \theta), \tag{1}$$

$$\left\{ \left( b_j^{'}, p_j^{'} \right) \right\} = M\left( x_u^{'}; \theta \right), \tag{2}$$

where $\{b_i\}$ in Equation (1) represents the bounding box information of the i-th predicted instance in image $x_u$, including the coordinate of the top left point (x1, y1) and the coordinate of the bottom right point (x2, y2). $\{p_i\}$ is the category probability distribution predicted by the model for the i-th instance, $p_i = \left[ p_i^1, p_i^2, \ldots, p_i^j, \ldots, p_i^n \right]$, where $p_i^j \in [0, 1]$ represents the probability that the current i-th instance is category $j$. The result $\left\{ \left( b_j^{'}, p_j^{'} \right) \right\}$ from Equation (2) is the prediction for $x_u^{'}$ and is not repeated.

#### 3.2.2. Predicted Box Matching

After model inference prediction, the prediction results of the original unlabeled image $x_u$ and its augmented view $x_u^{'}$ are attained. In this section, we will conduct predicted box matching to obtain paired predictions.

First, we perform transformation for prediction result $\{(b_i, p_i)\}$ of the original unlabeled image $x_u$:

$$\begin{cases} \hat{b}_i = A(b_i) \\ \hat{p}_i = p_i \end{cases}, \tag{3}$$

This is because, when the augmentation operated on the original image $x_u$ includes the horizontal flip or rotation of the image, the two groups of prediction results cannot be matched directly. It is necessary to perform the corresponding horizontal flip or rotation operation of the original image first to obtain the transformation result, so we can conduct the matching operation in the following.

Then, each instance $\left( \hat{b}_i, \hat{p}_i \right)$ is matched according to the maximum Intersection over Union (IoU) with $b_j^{'}$; the matching process can be formulated as follows:

$$b_i^{'} = \underset{b_j^{'} \in \{b_j^{'}\}}{\arg\max} \, IoU\left( \hat{b}_i, b_j^{'} \right), \tag{4}$$

From Equation (4), we can obtain the matching pairs $b_i^{'} \leftrightarrow \hat{b}_i$.

#### 3.2.3. Localization and Classification Stability Calculation

Specified for object detection tasks, we evaluate the stability of an instance from two aspects, namely localization stability and classification stability. Figure 2 shows the process of calculating the localization and classification stability of an instance. We select

the Intersection over Union (IoU), which can directly reflect the matching degree of two bounding boxes to measure an instance's localization stability.

$$ls_i = IoU\left(\hat{b}_i, b_i'\right), \tag{5}$$

where $ls_i \in [0, 1]$ represents the localization stability of predicted instance $b_i$. A high value indicates a stable and reliable instance localization prediction from the detection model.

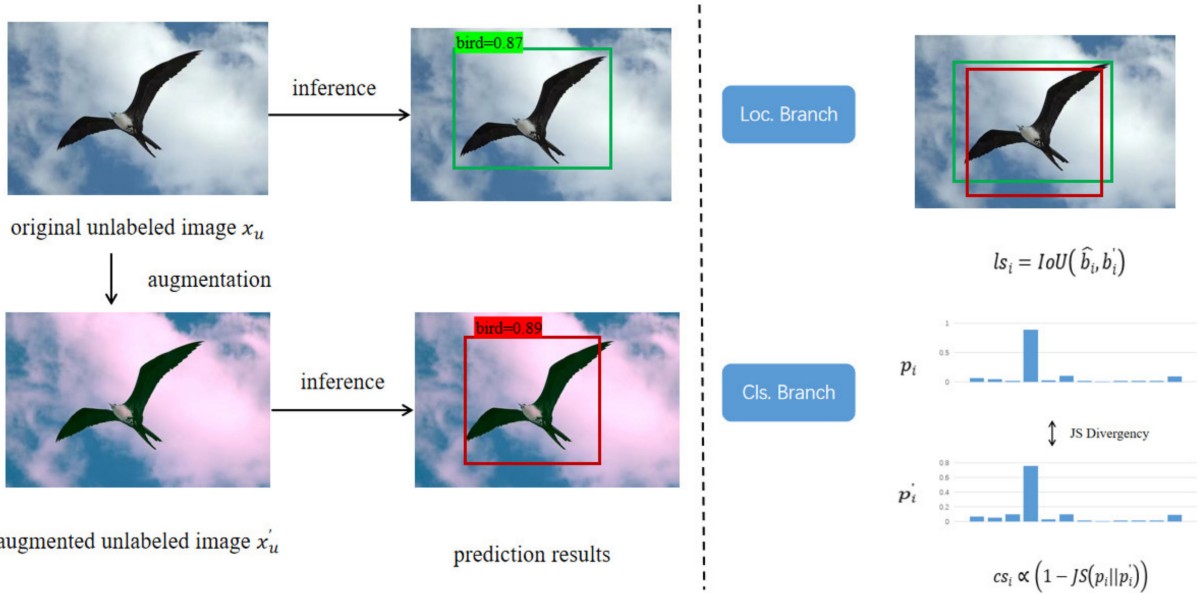

**Figure 2.** Localization and classification stability calculation process.

Jensen–Shannon (JS) divergence [41] is adopted to measure an instance's classification stability:

$$cs_i = \frac{1}{2}\left[\max_{p_i^n \in p_i} p_i^n + \max_{p_i'^n \in p_i'} p_i'^n\right]\left(1 - JS\left(p_i \middle\| p_i'\right)\right), \tag{6}$$

Jensen–Shannon (JS) divergence can describe the distance between two probability distributions, and has clear upper and lower bounds. The lower the JS divergence is, the closer two distributions are. 1—JS is used to reflect the stability of an instance. A high score indicates a stable and reliable instance classification prediction.

### 3.2.4. Determination of Sampling Metrics

Predicted instances can be categorized into four types based on the localization stability and classification stability calculated above. (1) The predicted instances with high localization stability $ls$ and high classification stability $cs$. As Figure 3a depicts (the green box indicates the predicted result from image $x_u$, and the red box indicates the predicted result from image $x_u'$; we place the two predicted results into the same image for better comparison), the detection model has high confidence in the category and precise location of the predicted instance. (2) The predicted instances with low localization stability $ls$ but high classification stability $cs$ (e.g., Figure 3b). The detection model is fairly confident that there is a person in the image, but cannot localize the person precisely. (3) The predicted instances with high localization stability $ls$ but low classification stability $cs$ (e.g., Figure 3c). The detection model can discover an object of interest in the middle of the image, but has trouble determining the category of the object. (4) The predicted instances with low localization stability $ls$ and low classification stability $cs$. Depicted in the left of Figure 3d, we can see that the predicted instances are most likely to be noisy samples when their ls and cs are low simultaneously.

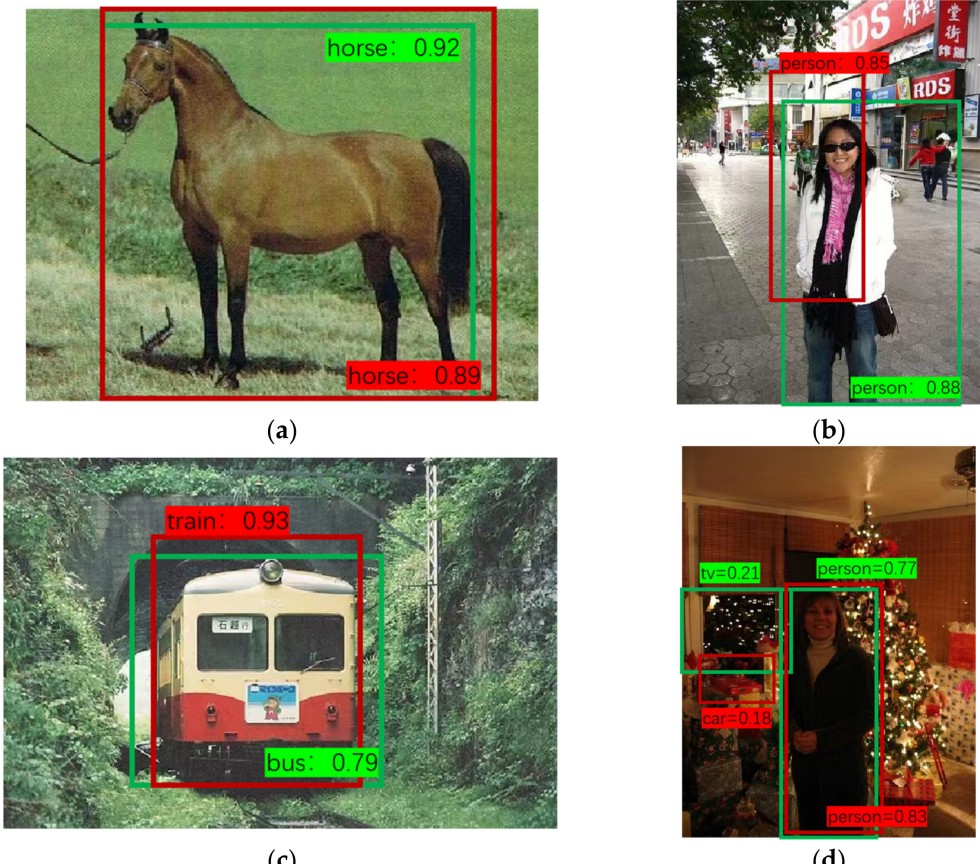

**Figure 3.** Example images for localization stability and classification stability analysis. (**a**) Example of high localization stability and high classification stability. (**b**) Example of low localization stability but high classification stability. (**c**) Example of high localization stability but low classification stability. (**d**) Example of low localization stability and low classification stability.

From the case analysis above, we obtain two types of informative predictions: (1) predictions with high classification stability and low localization stability; (2) predictions with low classification stability and high localization stability. The former indicates that the model is fairly certain of the class of instances but has difficulty in locating their specific locations, and therefore requires precise manual annotation; the latter indicates that the model is certain of the presence of its object at a certain location in the image but has difficulty in distinguishing its specific class, and therefore requires the assistance of manual annotation. Based on the observation, we design an instance's overall stability as

$$m_i = |ls_i + cs_i - 1|, \tag{7}$$

where $ls_i$ and $cs_i$ are the localization stability score and classification stability score of instance i calculated in Equations (5) and (6), respectively.

Thus far, we have formulated the definition of the stability of an instance. However, for images in object detection tasks, an image can consist of multiple instances. We select the minimum value of all stability values calculated in one image as the final stability score of the image, since object detection tasks focus on the most informative local region instead of the overall semantic information. For AL and SSL, the selected unlabeled samples are images rather than instances, so that a stability score for images is needed. The final stability score of an unlabeled image is formulated as

$$C = \min_i |ls_i + cs_i - 1|, \tag{8}$$

Thus far, we have completed the process of stability calculation. For unlabeled images with low stability, we use active learning to manually label them, and for images with high stability, we use semi-supervised learning methods for pseudo-label mining.

### 3.3. SSL Pseudo-Annotation Mining Strategy

To generate plausible pseudo-labels for the student model in semi-supervised object detection, it is common practice to apply a static threshold to filter out boxes with low confidence scores. Since the score of pseudo-labels can only indicate the confidence of pseudo-box categories, the localization quality of pseudo-boxes is not guaranteed. The static threshold also neglects the class imbalance problem existing in object detection, especially when annotations are scarce. Motivated by this, we introduce the stability aggregation score (SAS) and dynamic adaptive threshold (DAT), respectively.

### 3.3.1. Stability Aggregation Score

Considering that the classification score fails to indicate the localization quality, we introduce a simple yet effective method to measure the overall quality of pseudo-labels, named the stability aggregation score (SAS). For each unlabeled image $x_u$ in the unlabeled dataset, its augmented view is denoted as $x'_u = A(x_u)$; the original image $x_u$ and its augmented view $x'_u$ are simultaneously fed into the detection model M to obtain the corresponding prediction results $\{(b_i, p_i)\}$ and $\left\{ \left( b'_j, p'_j \right) \right\}$. Then, the matching between the prediction result pairs is performed according to the max IoU score to obtain the one-to-one corresponding prediction pairs $\left( \hat{b}_i, b'_i \right)$. The stability aggregation score of prediction instance i is formulated as

$$\text{SAC} = \begin{cases} \frac{1}{2}\left( max\hat{p}_i + maxp'_i \right) \cdot \text{IOU}\left( \hat{b}_i,\ b'_i \right), & \text{if } argmax\hat{p}_i = argmaxp'_i \\ 0, & \text{otherwise} \end{cases}, \tag{9}$$

From Equation (9), we can see that when the corresponding prediction results belong to the same category, the score of the instance prediction is determined by the confidence score indicating the classification status and the IoU score indicating the localization quality jointly. When the corresponding prediction results do not belong to the same category, it means that the category prediction results are very unstable, and for such a prediction instance, its stability score is directly set to 0. One can evaluate the overall quality of the predicted instance by combining the confidence score and the IoU score into the stability aggregation score.

### 3.3.2. Dynamic Adaptive Threshold

As discussed above, the static threshold bears the risk of amplifying the class imbalance problem. The accuracy of different categories often varies; those underrepresented classes usually come with relatively low confidence scores. Therefore, a static high threshold tends to filter out the lower-confidence predictions, amplifying the class imbalance effect. Simply lowering the threshold for this case will introduce very noisy pseudo-labels to other categories. Therefore, unlike the fixed thresholds commonly used in today's semi-supervised learning methods, in this paper, the thresholds for each category are dynamically adjusted according to the degree of stability of each category's localization and classification. For each category m, the corresponding threshold value is calculated as

$$\tau_m = \left( \frac{\sum_j ls_j^m \cdot cs_j^m}{E_m\left( \sum_j 1 \right)} \right)^\gamma \tau, \tag{10}$$

where $E_m\left( \sum_j 1 \right)$ in Equation (10) denotes the average number of foreground instances in all categories, and $\sum_j ls_j^m \cdot cs_j^m$ is the sum of the localization stability and classification stability

of the j-th instance of category m computed in the active learning sampling method. $\tau$ is an initial threshold set manually, and the factor $\tau$ is used to control the degree of attention to the underrepresented categories. When set to 0, the dynamic weighting factor is disabled and the formula degenerates to a fixed threshold. The thresholds are adaptively adjusted in real time based on the prediction results of the model by accumulating the locality stability and classification stability of all instances in each category online.

## 4. Experiments

### 4.1. Experimental Setup

We test the efficacy of our proposed method on PASCAL VOC [42]. The *trainval* sets of the VOC2007 and VOC2012 datasets, which contain 5011 and 11,540 images, respectively, are adopted for training. The detection performance is evaluated on the test set of VOC 2007, and mAP at an IoU of 0.5 is used as the evaluation metric.

We employ Faster-RCNN [43] with FPN [44] as our object detector and use ResNet-50 [45] as our feature extractor. We randomly select 5.0% labeled images from the training set to initialize the detector. In each learning cycle, the sampling method selects 2.5% images for manual annotation and 16% images for pseudo-labeling until the manually annotated images reach 20% of the training set. In each cycle, the model is trained for 20 epochs with the mini-batch size 4 and the learning rate 0.0025. The momentum and weight decay are set to 0.9 and $10^{-4}$, respectively.

### 4.2. Main Results

We compare our proposed method with state-of-the-art AL methods and SSL methods, respectively, to validate the superior performance of our proposed framework. For active learning, we compare the proposed method with random sampling, LL4AL [6], and SSM [13]. For semi-supervised learning, we compare the proposed method with CSD [46], STAC [37], Instant Teaching [39], and Unbiased Teacher [40]. For Instant Teaching, which uses ensemble techniques, we report its single-model results for a fair comparison.

Figure 4 shows the performance comparison with the state-of-the-art methods. The performance of all methods improves as the number of annotated images increases, indicating that manual annotation for unlabeled images can improve the performance of the detection model. With 20.0% annotation budgets, our method achieves 79.2% mAP, which significantly outperforms SSM by 4.3%.

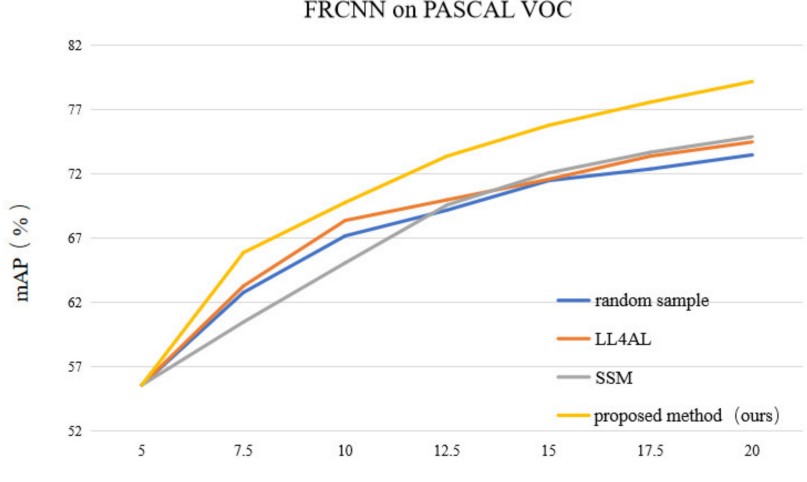

**Figure 4.** Performance comparison of active object detection methods.

For the SSL methods, the VOC07 *trainval* set is used as the labeled set and the VOC12 *trainval* set is used as the unlabeled set. The VOC07 test set is used for evaluation. From Table 1 we can see that the proposed method outperforms the exiting state-of-the-art SSL methods with even lower annotation budgets.

**Table 1.** Comparison with state-of-the-art SSL approaches on PASCAL VOC dataset.

| Method | Annotated Number | mAP |
|---|---|---|
| Supervised | 16,551 | 79.9 |
| CSD [46] | 5011 | 74.7 |
| STAC [37] | 5011 | 77.4 |
| Instant Teaching [39] | 5011 | 78.3 |
| Unbiased Teacher [40] | 5011 | 77.4 |
| Our Method | 3310 | 79.2 |

### 4.3. Ablation Study

We perform an ablation study on the key components of our proposed method. The study analyzes the impact on the detection performance of (1) our proposed active sampling strategy and (2) the proposed SSL pseudo-label mining method.

#### 4.3.1. Active Sampling Strategy

To validate the effectiveness of our proposed active sampling metric, we conduct an ablation study on different sampling strategies. We evaluate the detection performance on different sampling metrics. As depicted in Figure 5, the results show that localization stability or classification stability alone cannot represent the information contained in the images. Meanwhile, the evaluation results on ls+cs and |ls+cs-1| validate the effectiveness of the determination of the sampling metrics in Section 3.2. When we replace the min of |ls+cs-1| with the mean of |ls+cs-1|, the performance of the detection model drops significantly. This experimental result confirms our assumption: the detection tasks mainly focus on the informative region rather than the overall image.

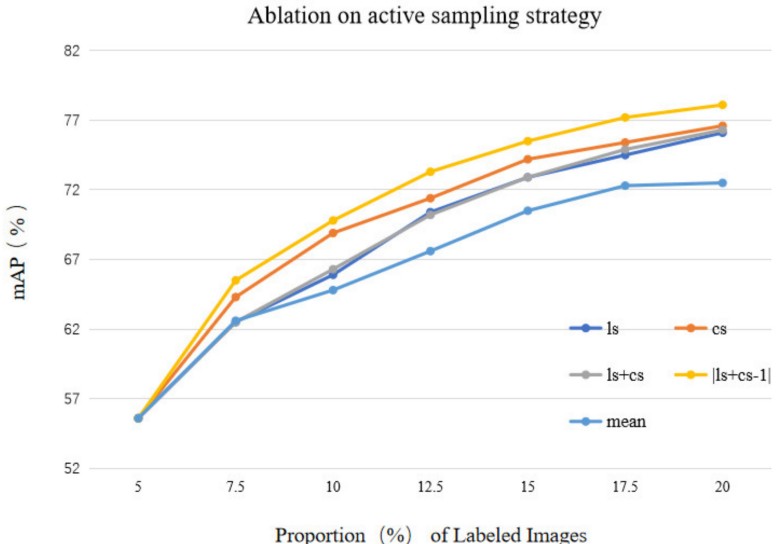

**Figure 5.** Ablation study on different active sampling strategies: ls only, cs only, ls + cs, |ls + cs − 1|, and mean strategy, respectively.

#### 4.3.2. SSL Pseudo-Label Mining Method

We further validate the effectiveness of our proposed SSL pseudo-label mining method, including the stability aggregation score (SAS) and the dynamic adaptive threshold (DAT). In Table 2, we show the effectiveness of each component step by step, where VOC0712

denotes the ensemble of datasets VOC2007 and VOC2012. When using 20% labeled data as the training samples in the regular SSL learning pattern, the model obtains 76.3% mAP. Next, we replace the confidence score with our proposed stability aggregation score, and the detection model obtains 78.6% mAP. Furthermore, by combining the SAS and ADT method into the framework, the detection model reaches 79.2% mAP. An improvement of +2.9% mAP is found compared to the original baseline.

**Table 2.** Comparison of mAP (%) for different methods.

| Methods | 10% VOC0712 | 15% VOC0712 | 20% VOC0712 |
|---|---|---|---|
| Confidence Score + Static Threshold | 70.3 | 74.0 | 76.3 |
| SAS + Static Threshold | 70.4 | 75.7 | 78.6 |
| SAS + DAT | 70.3 | 76.1 | 79.2 |

## 5. Discussion

In this section, some limitations of our proposed approach are discussed. First of all, in the active sampling process, the lowest stability instance score is used as the stability score of the entire unlabeled image. However, an unlabeled image from the unlabeled dataset typically contains multiple instances, and the active sampling process must provide manual labeling for all instances appearing in the unlabeled image. Among all annotated instances, only one unstable instance is needed, which may result in a waste of labeling resources. Secondly, in the semi-supervised learning process, although the incorporation of SAS and DAT ensures the quality of the pseudo-labels, there are still several hard samples that cannot be successfully labeled by the pseudo-annotation mining strategy, resulting in the leakage of labels. Figure 6 shows some failure cases in the SSL pseudo-labeling process. The green rectangles denote successful pseudo-labels, and the yellow rectangle denotes the ground truth box, but the SSL pseudo-annotation mining strategy fails to detect them.

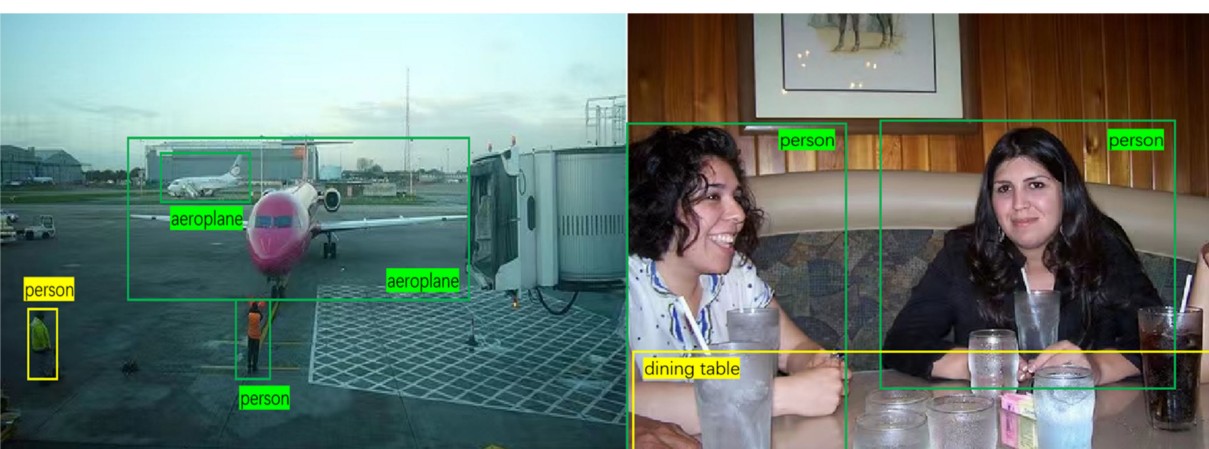

**Figure 6.** An illustration of some failure cases. The green rectangles denote successful pseudo-labels, and the yellow rectangle denotes the ground truth box, but the SSL pseudo-annotation mining strategy fails to detect them.

## 6. Conclusions

In this work, we propose a semi-supervised active learning framework tailored for object detection. Our proposed method can automatically discover potential instances from an unlabeled image. Customized for object detection tasks, we evaluate the stability of an instance from two aspects, namely classification stability and localization stability. For highly stable instances, we consider them as high-quality instances and pseudo-label them with predicted classification and regression information. For those relatively unstable

instances, we manually annotate them to ensure that we do not introduce noisy samples, which can only degrade the model's performance. We also extensively analyze the existing pseudo-label mining strategy for SSOD and observe that these strategies currently neglect some important properties of object detection. Motivated by this, we propose a new pseudo-label mining strategy consisting of SAS and DAT. Experimental results confirm that our framework surpasses the SOTA by a large margin. On the PASCAL VOC benchmarks, our proposed method can improve the 74.7% mAP at baseline to 79.2% mAP, surpassing previous methods by approximately 1% mAP, with fewer annotated images.

From the analysis in the Discussion section, we can see that object detection is an instance-level task, and operations at the image level may cause a waste of labeling resources and the leakage of hard labels. While our proposed method demonstrates an impressive performance gain, further investigation of instance-level incorporation could be a promising direction to further enhance the performance of the object detection framework.

**Author Contributions:** Conceptualization, S.C.; methodology, S.C. and Y.Y.; software, S.C.; validation, S.C., Y.Y. and Y.H.; formal analysis, S.C.; investigation, S.C.; resources, Y.Y.; data curation, S.C.; writing—original draft preparation, S.C.; writing—review and editing, S.C., Y.Y. and Y.H.; visualization, S.C.; supervision, Y.Y.; project administration, Y.Y. and Y.H.; funding acquisition, Y.Y. All authors have read and agreed to the published version of the manuscript.

**Funding:** This research was funded by the National Key R&D Program of China, grant number 2020YFB1406800.

**Data Availability Statement:** The Pascal VOC dataset used in this paper can be downloaded from the link http://host.robots.ox.ac.uk/pascal/VOC/, accessed on 1 January 2020.

**Conflicts of Interest:** The authors declare no conflict of interest.

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
