# Peer review of "Semi-Supervised Active Learning for Object Detection"

_electronics, doi:10.3390/electronics12020375_

Round 1

Reviewer 1 Report

The article considers active learning and semi-supervised learning for solving object detection. Based on the uncertainty of an unlabeled image, the image will be either labeled through active learning, or treated as pseudo-labeled training samples in SSL. The method show promising performance on standard benchmarks. However, several issues should be addressed:

The writing of the paper can be improved. For example, in line 124, the meanings of (x1,y1) and (x2,y2) are not clear. In line 128, the symbols X_t^al and X_t^ssl are not defined as well.

One question for Eq.7 and Eq.8 is why a stability score is defined at the image level in Eq.8, rather than at the instance level.

The literature review is not comprehensive. Active learning has been widely explored in vision tasks, like Volumetric memory network for interactive medical image segmentation for segmentation, and Weakly Supervised 3D Object Detection from Lidar Point Cloud for object detection. They should be included in the paper.

The limitations of the method should be discussed. It would be better to show some failure cases to gain more insights about the approach.

Author Response

We appreciate the reviewers for the thorough review of our submission and believe that our paper has benefited from their insightful comments. According to the reviewers’ comments, we have made some revisions of the manuscript. The revisions and responses to reviewers are marked in color. Please see the attachment. 

Reviewer 2 Report

The paper proposed a semi-supervised active learning framework for object detection. The paper structure is fine as well as the writing. In addition, the method performance has been evaluated on Pascal VOC dataset and satisfactory results have been reached. However, some minor revisions have to be done such as:

1- make figure 1 and figure 3 centered.

2- in the abstract : add a space before "(SAS)" and "(DAT)"

3- Add more keywords (no less than 5-6 keywords in total)

Author Response

We appreciate the reviewers for the thorough review of our submission and believe that our paper has benefited from their insightful comments. According to the reviewers’ comments, we have made some revisions of the manuscript. The revisions and responses to reviewers are marked in red. Please see the attachment. 

Reviewer 3 Report

This paper presents an object detection framework that combines active learning and semi-supervised learning.

The author shouldn't mention too many publications in bulk, for instance, the author cites references [10-21] at line 33, 49, 51, 64, 71. Some clarifications were required here. 

The presentation of paragraph 2.1 is poor. It is uncommon for a paragraph to begin with a figure. Some equations are presented in this paragraph. There are no numbers to these equations.

The model inference prediction is presented in paragraph 2.2.1. (1) and (2) must be referred to or demonstrated. The same observation for (4), (6), (9) and (10). 

The discussion section could be revised based on future scope and numerical results.

Author Response

(The authors gave the same response as above.)

Round 2

Reviewer 1 Report

The revision looks promising. I have no further concerns.